# Peridynamics for Fracture Analysis of Reflective Cracks in Semi-Rigid Base Asphalt Pavement

**Zhichuang Shi [1], Jinchao Yue [1], Lingling Xu [2,* and Xiaofeng Wang [3]**

1   School of Water Conservancy Engineering, Zhengzhou University, Zhengzhou 450001, China; scc2511@gs.zzu.edu.cn (Z.S.); yuejc@zzu.edu.cn (J.Y.)
2   College of Civil Engineering, Henan University of Technology, Zhengzhou 450001, China
3   Henan Provincial Communications Planning & Design Institute Co., Ltd., Zhengzhou 450001, China; xfyd315@sohu.com
*   Correspondence: xllfyc@163.com

**Abstract:** Reflective cracking is one of the major forms of deterioration in semi-rigid base asphalt pavements. It is, therefore, very important to have a correct understanding of the internal crack propagation mechanism of asphalt pavement to propose the most effective remedial solution(s), which corresponds to that mode of failure. In this study, two-dimensional asphalt pavement layered models are first established by modifying the peridynamics theory. Then, the influence of asphalt overlay thickness and load form on reflective crack propagation is explored. On this basis, the influence of friction between the tire and road surface on reflective crack propagation is analyzed. The results show that increasing the thickness of the asphalt overlay can inhibit reflective crack propagation, and the friction force accelerates reflective crack propagation when the direction of friction is the same as that of reflective crack propagation; otherwise, it inhibits reflective crack propagation. Additionally, the most unfavorable load position is the asymmetrical load when the vehicle is far from the reflective crack.

**Keywords:** peridynamics; layered model; reflective crack; crack propagation; structure failure

## 1. Introduction

Asphalt pavement with a semi-rigid base is a widely used structure for high-grade highways due to a number of advantages, including high strength and stiffness, sound flatness, and good fatigue performance. However, this type of base course easily produces dry shrinkage cracks and low-temperature shrinkage cracks during its strength formation because of its sensitivity to changes in humidity and temperature. The repeated action of traffic load will lead to stress concentration at the crack tip, and further cracking will soon form and even propagate up to the flexible layer. Such a bottom-up crack is called a reflective crack. Reflective cracks by themselves may not have a serious impact on the structural bearing capacity of composite pavement. However, water infiltration through the reflective cracks at the joints, together with the heavy traffic movements, can accelerate structural pavement failure, greatly weakening the strength and stability of the subgrade, leading to early failure of the pavement and reducing the life of the pavement. Therefore, research on the mechanism of cracking growth will help monitor the cracks' behaviors from the beginning of their development and offer feasible methods manufacturing and maintenance.

Many studies have been carried out to investigate cracking mechanisms [1–4]. Conventional research on crack behavior relies heavily on laboratory-scale experiments, which require considerable time and effort. Molenaar [5] and Millien et al. [6] found that reflective cracks are produced under the joint action of temperature and traffic through a large number of experimental studies. Chen et al. investigated the premature cracking distress in pavement by field tests that also involved destructive tests [7]. These experiments always focused on the final status of cracks and pavement, and the processes of crack growth were

ignored due to testing methods. Over these years, advancements in computer technology have allowed realistic simulations of the performance of materials to be performed. The finite element method (FEM) has become a popular approach to simulate crack behavior during their life cycle. Myers [3] established a two-dimensional plane strain model using the finite element software ABAQUS. The settlement results reveal the propagation law and direction of precast cracks under traffic loads, which provides a mature method for pavement simulation. García [4] used the method of preset interlayer cracks to predict the cracking of each layer of pavement during the use. The results show that the initial cracks mostly occur at the joints between layers, the free boundaries of each layer, and the thinner layers. Alae [8] found that top-down cracks in pavement structures can be broken down into I + II fractures by establishing a three-layer 3D model of asphalt pavement with top-down cracks, in which the working temperature and vehicle speed were assumed to be the changing working conditions of the pavement structure. Parameters of crack growth and their time history curve can be calculated using the finite element method and predictions of pavement life, and the degree of damage can be drawn. However, the finite element method is based on partial differential equations (PDEs) within the framework of classical continuum mechanics. All these methods within the scope of classical continuum mechanics suffer from the inherent limitation that the spatial derivatives required by the PDEs, by definition, may lose their meaning due to the presence of a discontinuity, such as a crack.

To overcome this limitation, many special techniques within the framework of classical continuum mechanics have been developed, such as the cohesive zone model (CZM) [9] and the extended finite element method (XFEM) [10]. All these methods redefine a body in such a way as to exclude the crack and then apply conditions at the crack surfaces as boundary conditions. Li et al. [11] employed XFEM to investigate the fracture process of dowel bars in rigid pavement. Lancaster et al. [12] developed an XFEM model of a semicircular bending test to simulate crack propagation and compared it with a laboratory semicircular bending test. Wang et al. [13–15] studied the reflective crack propagation of asphalt pavement under single temperature and circle temperature loads and studied the reflective crack propagation under the action of temperature–traffic coupling dynamics with XFEM. However, within these methods, the continuity and the discontinuity of a solid body are described by different governing equations, which means external criteria are needed to define the onset of crack growth, the direction, and the speed of crack propagation, and even to answer whether the crack should turn, branch, oscillate, arrest, etc.

Peridynamics (PD), proposed by Silling et al. [16–18], is a nonlocal theory of mechanics that replaces spatial derivatives with an integral operator and introduces a new definition for damage. The PD theory avoids singularities at the crack tip because it introduces a length-scale, the size of the nonlocal region. Meshfree discretization methods for PD models (based on midpoint integration of the integral operator) can simulate the natural evolution of damage and fracture seamlessly. Such models have natural theoretical advantages in analyzing crack propagation and severe damage of solids under dynamic loading such as impact [19–21], in which the crack nucleation and crack propagation are spontaneously simulated by applying a simple bond damage criterion without any additional criterion for judging whether the crack propagates. PD theory has been widely used by scholars to analyze the fracture of different types of material. Gu et al. [22] studied the elastic wave dispersion, propagation, and impact damage of the concrete Brazilian disc in the split Hopkinson pressure bar (SHPB) test based on a PMB (Prototype Microelastic Brittle) peridynamics model. Pan Wu et al. [23] applied an intermediately homogenized peridynamics (IH-PD) model of concrete to numerically investigate the dynamic failure behavior in concrete. Peng et al. [24] established a micro-calculation model of concrete by the MATLAB-ABAQUS co-simulation method based on the PD theory, and the evolution law of fracture reflects the micro-characteristics and macro-evolution law of fracture well. Liu et al. [25] established a numerical model of asphalt mixtures based on PD theory, analyzed the stress state and energy changes in specimens during the semicircular bending test, and analyzed

the effects of coarse aggregates, voids, and microcracks on the low-temperature cracking of asphalt mixtures by using a two-dimensional micronumerical model. Ruan et al. [26,27] used the numerical method of PD theory to simulate the crack branching stage of a cylindrical asphalt mixture used in the laboratory under a compression load. In view of the advantages of PD theory in dealing with discontinuous problems, many scholars have devoted themselves to the study of PD theory in recent years, and PD theory has good development prospects. However, there are few studies on semi-rigid multilayer pavement based on this theory.

In this paper, the theory of peridynamics is briefly described, and the peridynamics algorithm program is compiled by MATLAB. After that, the correctness of the written program is verified by several typical examples. Then, the pavement model of the multilayer structure is built, the crack propagation trajectory of different positions under load is obtained through simulation calculation, and the influence of loads at different positions on the expansion of existing reflective cracks in the semi-rigid base is analyzed. On this basis, the calculation model is simplified to simulate the reflective crack propagation under different thicknesses of asphalt overlay, and the influence of asphalt overlay on reflective crack cracking is analyzed. Finally, considering the friction between the vehicle tire and asphalt pavement, the most unfavorable load position is obtained by comparing the crack propagation under different partial load positions.

## 2. Peridynamics

### 2.1. Peridynamics Theory

PD theory assumes that an object occupies a space domain R, and at a certain moment t, there is an interaction between any material point x in the space and other material points x' within a certain radius of the surrounding space, which is called the integral domain, also the horizon. Such interaction is called a bond, that is, the force between points is transferred through the bond, as illustrated in Figure 1. If the interaction force is f, the motion state of a material point is the result of the joint action of the external force and all the material points within the range of its space domain. Then,

$$\rho \ddot{u}(x) = \int_{H_x} f\big(x,\, x',\, u(x,\, t),\, u(x',\, t)\big) dV + b(x,\, t) \tag{1}$$

where $\rho$ is the mass density, u is the displacement vector, $\ddot{u}$ is the acceleration, b is the external applied force density, and $H_x$ is the horizon or the interaction domain of a material point located at position x, which can be expressed by the following formula:

$$H_x = H(x,\, \delta) = \big\{ x' - x \leq \delta \big| x' \in \delta \big\} \tag{2}$$

where $\delta$ is the horizon region.

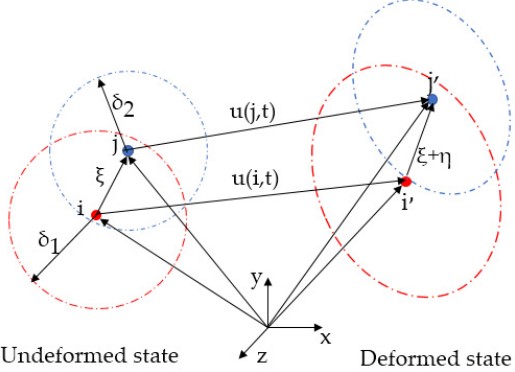

**Figure 1.** Interaction between material points.

The relative position vector of the PD point is:

$$\xi = x' - x \tag{3}$$

The relative displacement vector in the deformed configuration at time t is:

$$\eta = u(x', t) - u(x, t) \tag{4}$$

Then, the expression of the force between PD points can be simplified as:

$$f = f(x, x', u(x, t), u(x', t), t) = f(\eta, \xi) \tag{5}$$

For linear elastic materials, f is the derivative of the microelastic strain density $\omega$ with respect to the relative displacement vector $\eta$, that is:

$$f(\eta, \xi) = \frac{\partial \omega(\eta, \xi)}{\partial \eta} \; \forall \eta, \; \omega \tag{6}$$

The microscopic elastic strain energy of the bond is:

$$\omega(\eta, \; \xi) = \frac{cs^2}{2}|\xi| \tag{7}$$

where the micromodulus parameter c is the stiffness of the bond, $|\xi|$ denotes the magnitude of vector $\xi$, and the scalar bond stretch is defined as:

$$s = \frac{|\eta + \xi| - |\xi|}{|\xi|} \tag{8}$$

Therefore, the constitutive function of the PD model can be obtained:

$$f = \begin{cases} cs\mu\frac{\xi+\eta}{|\xi+\eta|} , |\xi| \leq \delta \\ 0, |\xi| > \delta \end{cases} \tag{9}$$

where c is the micromodulus and $\mu$ is the characteristic function of the bonding state. $|\xi + \eta|$ and $|\xi|$ are the deformed length and initial length of the bond, respectively.

### 2.2. The Constitutive Relation

Similar to traditional theory, the PD constitutive relation is also the relation between the state of material force and the state of deformation, and the material parameters in the PD constitutive relation can be deduced from the material parameters in the traditional constitutive relation. The expression of micromodulus c and the value of Poisson's ratio $\nu$ in PD can be obtained in 2D and 3D, as displayed in Table 1.

**Table 1.** Micromodulus c in BPD.

| | Expression of the Micromodulus | Poisson's Ratio | Micromodulus |
|---|---|---|---|
| Plane strain (2D) | $\frac{6E}{\pi h \delta^3 (1+\nu)(1-2\nu)}$ | $\nu = \frac{1}{3}$ | $c = \frac{48E}{\pi h \delta^3}$ |
| Plane stress (2D) | $\frac{6E}{\pi h \delta^3 (1-\nu)}$ | $\nu = \frac{1}{3}$ | $c = \frac{9E}{\pi h \delta^3}$ |
| 3D | $\frac{6E}{\pi \delta^4 (1-2\nu)}$ | $\nu = \frac{1}{4}$ | $c = \frac{12E}{\pi \delta^4}$ |

where E and h are Young's modulus and thickness for the 2D case.

In the traditional constitutive model, for elastic-brittle materials, the material suddenly fails when the stress reaches the peak value. Therefore, the characteristic function $\mu$ in the PD constitutive function can be defined as:

$$\mu(\eta,\ \xi,\ t) = \begin{cases} 1, & (s < s_0) \\ 0, & (s \geq s_0) \end{cases} \tag{10}$$

where $s_0$ is the ultimate elongation, that is, the critical deformation value of bond fracture. When the deformation of the bond exceeds the critical deformation value, it breaks and no longer transmits load.

For the two-dimensional plane stress problem, the ultimate elongation of the bond is:

$$s_0 = \sqrt{\frac{4\pi G_0}{9E\delta}} \tag{11}$$

where $G_0$ is the material's critical fracture energy.

Therefore, nonlocal damage parameters of material points can be defined by bond fracture, that is, to characterize the damage of material points:

$$D(x,t) = 1 - \frac{\int_{H_x} \mu(\eta,\delta,t)dV}{\int_{H_x} dV} \tag{12}$$

The range of the material point damage degree is $0 \leq D(x,t) \leq 1$, where 0 represents the intact material point without damage, and 1 represents the complete failure of the interaction bond around the material point. It can be seen that the PD method has its own failure criterion, and there is no need to introduce other criteria to judge whether the material is damaged.

### 2.3. Numerical Implementation

In the reference construction, the computational domain is discretized into points containing a given volume. In the discrete formula for equation of motion (1), the original integral calculation is replaced by the form of finite term summation:

$$\rho\ddot{u}_i^n = \sum_p f(u_p^n - u_i^n, x_p - x_i)V_p + b_i^n \tag{13}$$

where f can be calculated by formula (9), $V_p$ is the volume occupied by points P, and n is the time iteration step. The subscript indicates the point number:

$$u_i^n = u(x_i,\ t^n) \tag{14}$$

The motion equation of the PD model is expressed as:

$$\rho\ddot{u}_i^n = \sum_p C(x_p - x_i)(u_p^n - u_j^n)V_p + b_i^n \tag{15}$$

The motion equation of the PD model is expressed as:

$$\ddot{u}_i^n = \frac{u_i^{n+1} - 2u_i^n + u_i^{n-1}}{\Delta t^2} \tag{16}$$

where $\Delta t$ is a constant time step.

### 2.4. PD Model of the Layered Structure

PD theory discretizes objects into points of matter characterized by specific material properties. For homogeneous materials, only uniform discretization is needed. The layered structure must undergo special treatment because the material properties vary from layer to layer. A layered structure is a type of nonuniform material in nature. For the convenience of analysis, the layered structure is regarded as a composite material that is uniform within

the layer but not uniform as a whole. Here, uniformity in a single layer means that elastic parameters such as elastic modulus, density, and fracture energy are assumed to be constant at any material point in a layer structure and in a small enough area nearby.

Consider any pair of material points x and y connected by bonds between layers in a layered structure, whose effective elastic modulus, density, and fracture energy are $E_x$, $\rho_x$, $G_x$ and $E_y$, $\rho_y$, and $G_y$, respectively. In the PD model, it is necessary to obtain the micromodulus between material point pairs. Here, the average value of elastic parameters of the material point to X and Y is used to calculate the bonding micromodulus between them:

$$E(x,\ y) = \frac{E_x + E_y}{2} \tag{17}$$

$$G(x,\ y) = \frac{G_x + G_y}{2} \tag{18}$$

Since material density is the parameter of the material point, the density of each point can be given according to the distribution model of the gradient material in the PD model.

## 3. Numerical Model Validation
### 3.1. Cantilever Beam under Uniform Tension

In this section, the PD numerical model of the cantilever beam is established, and the simulation results are compared with the analytical solution to verify the accuracy of the program in solving the linear elastic static problem.

The geometric dimensions of the cantilever beam are as follows: the length is 100 mm, the width is 20 mm, and the thickness is 1 mm. The boundary constraints of the model are shown in Figure 2. The elastic modulus of the material is E = 100 GPa, Poisson's ratio is v = 1/3, and the uniform pressure $P_0$ = 100 MPa is applied to the right end of the model. The cantilever beam was discretized by a uniform distribution of material points with a spacing of $\Delta$ = 1.0 mm, and the material horizon was selected as $\delta$ = 3.0$\Delta$, as proposed in many PD applications. According to the theory of elasticity, the theoretical solution at each position of the model under this load is:

$$u_x(x,\ y) = \frac{P_0}{E}x \tag{19}$$

$$u_y(x,\ y) = -v\frac{P_0}{E}y \tag{20}$$

where $u_x$ and $u_y$ are the displacements along the x direction and y direction, respectively, and v is the Poisson's ratio of the material.

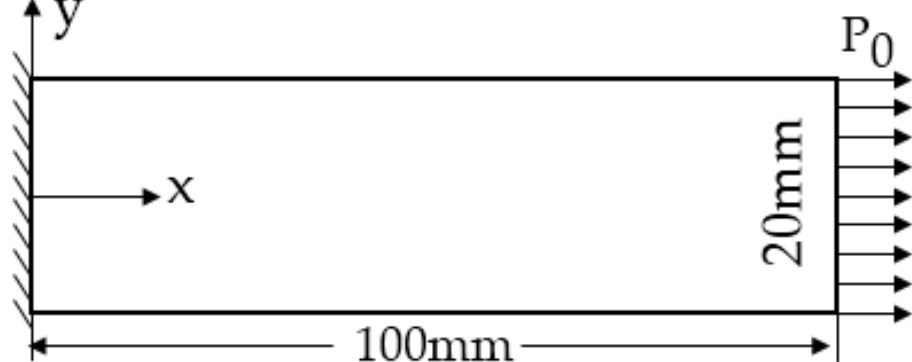

**Figure 2.** Tensile model of cantilever beam.

The calculated results of displacement $u_x$ along the x direction of the point on the line y = 0 are compared with their analytical solutions in Figure 3. The comparison results show

that the PD solution is in good agreement with the theoretical solution. This also indicates that the above discrete method is effective and reliable.

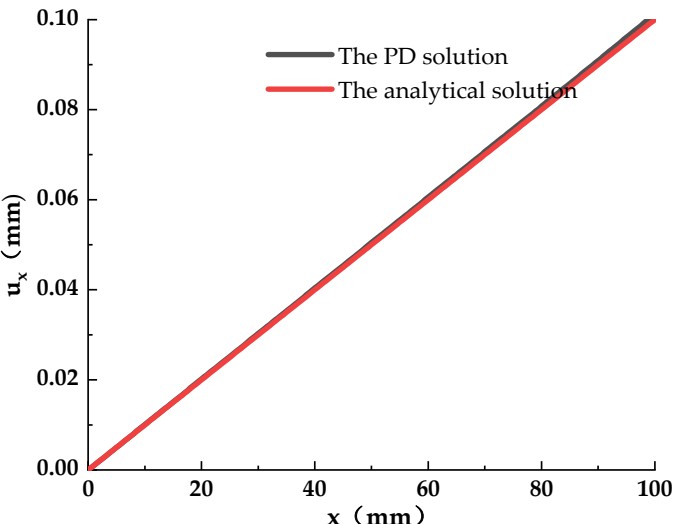

**Figure 3.** Comparison of displacement curves.

### 3.2. Dynamic Crack Branching Verification

Crack branching is a very complex process in the process of crack propagation. Ruan et al. [27] simulated the dynamic crack branching phenomenon of brittle materials by coupling the lattice bond element method. PD theory can simulate crack bifurcation naturally without external criteria. Then, a numerical example is used to establish a PD model to simulate crack bifurcation, and the accuracy of crack propagation and bifurcation is verified by existing experiments.

As shown in Figure 4, there is a unilateral pre-existing crack with a length of 50 mm in the middle of the rock material. The length of the rock plate is L = 100 mm, the width is W = 40 mm, the elastic modulus is E= 72 GPa, the material's critical fracture energy is $G_0$ = 135 J/m$^2$, the density is ρ = 2650 kg/m3, and the plate is subjected to upper and lower tensile loads of $\sigma_0$ = 12 MPa. All physical parameters, the size of the pre-existing crack, the size of the load, and the position of the application in this example are the same as those in [28].

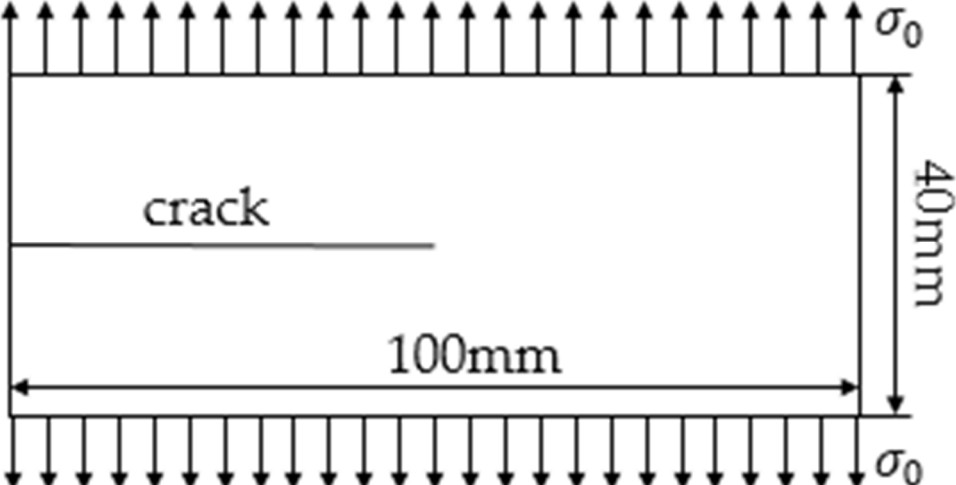

**Figure 4.** Model dimensions and load vector diagram.

Figure 5 shows the crack growth path under 1.2 and 12 MPa, which are consistent with the experimental results of Bobaru [28], proving that the proposed method can predict the crack growth of the model under different loads.

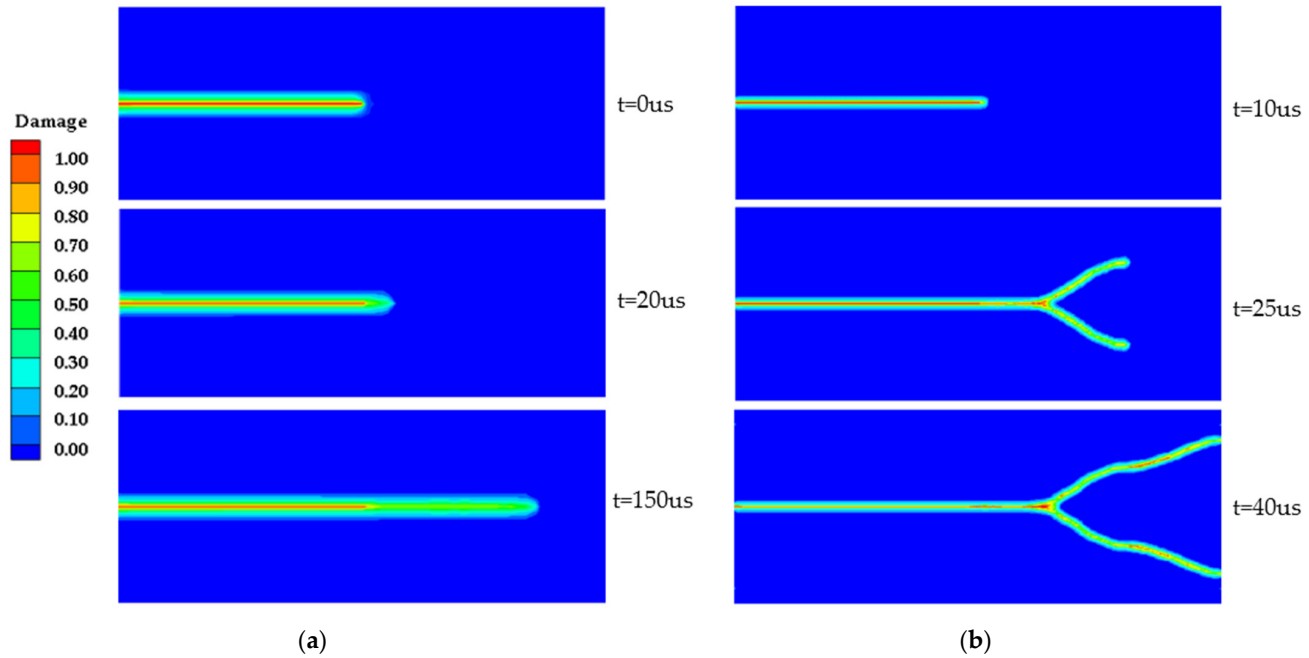

(**a**)                                                                                          (**b**)

**Figure 5.** Crack propagation path diagram. (**a**) Crack propagation at 1.2 MPa; (**b**) crack propagation at 12 MPa.

### 3.3. Multilayer Structure Model

Semi-rigid base asphalt pavement can be simplified to a multilayer structure. In the first two sections of this chapter, the correctness of the PD algorithm applied to the homogeneous beam and homogeneous plate cracking process was verified. In this section, the PD algorithm program was extended to apply it to a multilayer structure, and the simulation results were compared with the simulation results of finite element software to verify its correctness.

As shown in Figure 6, the size of the multilayer structure simulation model is 100 × 60 mm, the bottom of the model is fixed, and the top is subjected to a uniform load of P = 100 MPa. The detailed material parameters of each layer of the model are shown in Table 2.

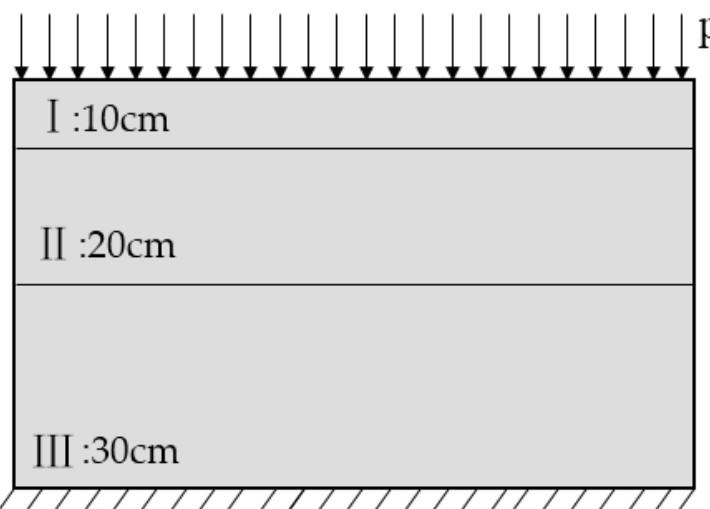

**Figure 6.** Multilayer structure model.

**Table 2.** Material parameters.

|  | Elastic Modulus | Poisson's Ratio |
|---|---|---|
| I | 2000 MPa | 1/3 |
| II | 1500 MPa | 1/3 |
| III | 1300 MPa | 1/3 |

The simulation model of the multilayer composite structure is established by PD theory, which is separated into 101 points in the X direction and 61 points in the Y direction with a discrete distance of 1 mm, and the material horizon is $\delta = 3.0\Delta$. At the same time, the finite element software Abaqus was used to establish the same simulation model as the control group, and the mesh size was controlled to be the same as the discrete spacing of the PD points, which was 1 mm. The calculation results of the PD algorithm and finite element algorithm are compared, as shown in Figure 7. The results show that the calculation results of the two algorithms are in good agreement, indicating that the proposed layered models are effective and reliable.

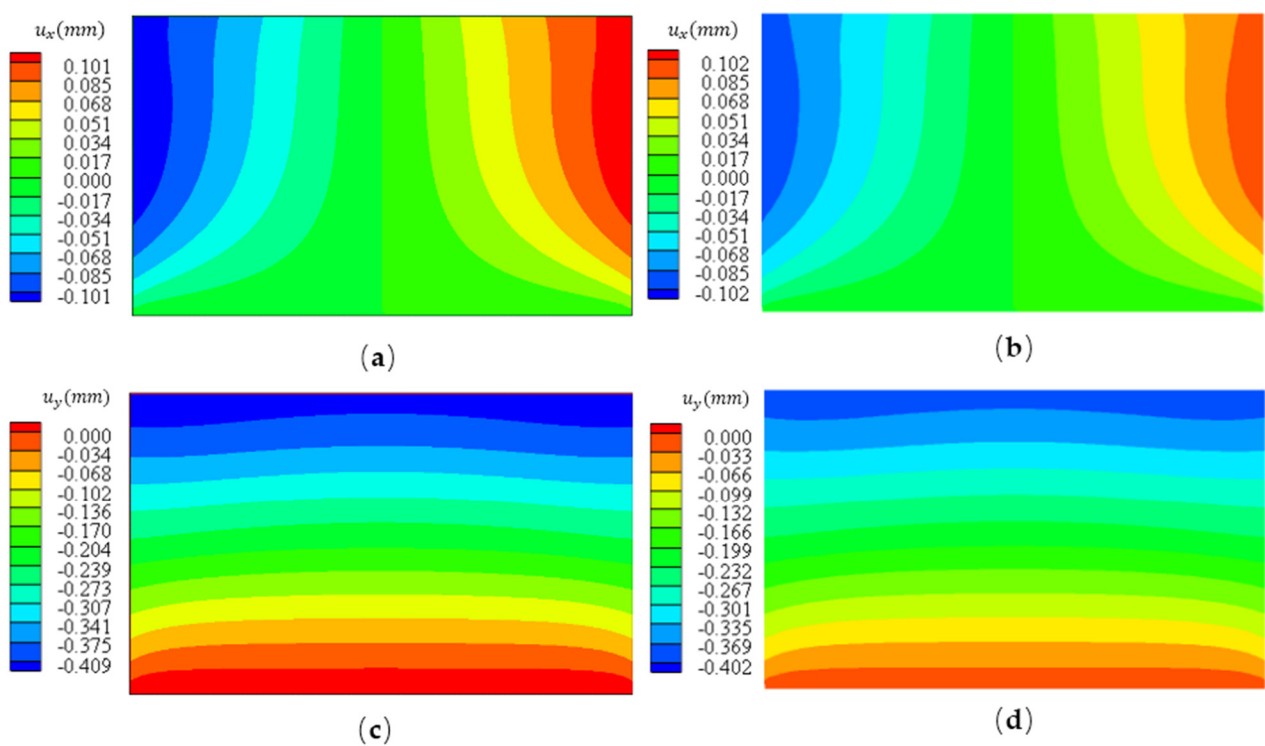

**Figure 7.** Comparison of displacement contours by using FEM and PD (unit: mm). (**a**) Horizontal displacement by FEM; (**b**) horizontal displacement by PD; (**c**) vertical displacement by FEM; (**d**) vertical displacement by PD.

## 4. Pavement Simulation Model

Semi-rigid base asphalt pavement has a multilayer structure. In many studies, these layers are generally considered to be perfectly bonded or slip [29–32]. Ninety percent of the low-temperature crack resistance of asphalt pavement is provided by the asphalt overlay. Asphalt becomes hard and brittle at low temperature, which is one of the main reasons for the low-temperature cracking of asphalt pavement. An asphalt mixture is a heterogeneous anisotropic material composed of aggregates, asphalt, and voids, but an asphalt mixture can be approximated as a microelastic solid at low temperature [25]. Therefore, the application of PD theory to the simulation of low-temperature cracking of asphalt pavement has a scientific, feasible, and sufficient research basis [25–27].

The analysis process of numerical simulation involves using the model established by software to simulate the mechanical behavior of semi-rigid base pavement, among which the most influential factors are the correctness of the model and the determination of boundary conditions. For the semi-rigid base pavement structure model, the following assumptions are made:

1.  Each structural layer of the road surface is a homogeneous and linear elastomer;
2.  The contact condition between pavement structure layers is completely continuous;
3.  There is no load transfer capacity between cracks;
4.  The deformation of the pavement structure is small, and the gravity of each structural layer is not considered.

### 4.1. Constitutive Relationship of Asphalt Mixture

Reflective cracks are formed by shrinkage cracks in semi-rigid bases under the repeated action of traffic loads. Therefore, this paper mainly studied the cracking of asphalt overlays. An asphalt mixture under low temperature is regarded as an ideal elastoplastic material. The asphalt mixture has good compressive performance at $0°$ but poor tensile performance. Therefore, tensile and compressive properties should be considered separately. In combination with the performance parameters measured in the experimental part [32,33] and Shen et al.'s [34] research on the PD model of concrete, this paper proposes a near-field dynamics homogenization model at low temperature. The relationship between the bond forces and expansion rate of material points is shown in Figure 8.

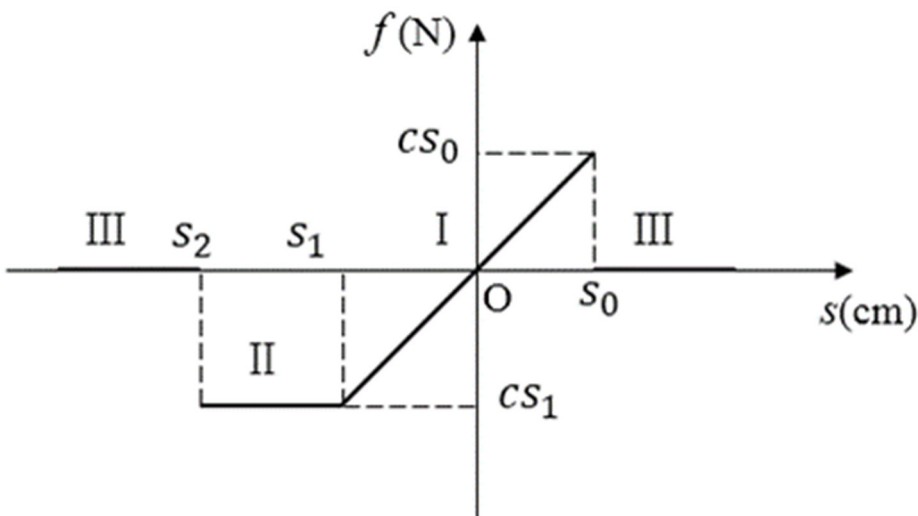

**Figure 8.** Constitutive model of asphalt mixture.

$s > 0$ is the tension zone of the material, while $s > 0$ is the compression zone. In elastic stage I, the "bond" between the material point and other points in the near field is not broken, and the interaction force between the two material points changes linearly with the change in elongation. In plastic stage II, the material reaches a plastic state, the "bond" between material points is not broken, but the force between material points does not increase when the elongation increases. In stage III, the "bond" between material points is destroyed, the interaction force falls to zero, and the broken "bond" cannot be recovered. The interaction force function of PD at each stage can be expressed as follows:

$$f = c \cdot s \quad s_1 \leq s \leq s_0 \tag{21}$$

$$f = c \cdot s_1 \quad s_2 \leq s \leq s_1 \tag{22}$$

$$f = 0 \quad s < s_2 \text{ or } s > s_0 \tag{23}$$

where c and $s_0$ are calculated by using the parameters of Table 1 and formula (11), respectively. $s_1$ and $s_2$ are calculated according to the following formulas in reference to Gerstle et al. [35] and Nicolas's [36] research.

$$s_1 = \frac{f_c}{E_c} \quad s_2 = n \cdot \frac{f_c}{E_c} \tag{24}$$

where $E_c$ is the elastic modulus of the asphalt mixture at 0 °C and $f_c$ is its compressive strength. As the asphalt mixture is rarely damaged under compression, the plastic deformation space is large, so n = 10. By referring to formula (11) and setting $s_0 = s_1$, the relationship between the energy release rate $G_0$ and compressive strength $f_c$ can be solved.

### 4.2. Pavement Structure Design

Whether the model size definition is reasonable or not has a great influence on the accuracy of the simulation. If the model size is large enough, the real effect of the simulated structure force is better. However, an excessively large size will affect the calculation speed, so the calculation accuracy and calculation quantity need to be considered comprehensively. Previous studies show that a 6 × 6 × 5 m cuboid specimen was the most suitable type for simulation of the actual pavement structure [37,38]. In this paper, a two-dimensional PD model of a pavement structure is established, and no lateral movement is selected as the boundary condition to reveal the actual situation of crack propagation. Considering that the displacement of the subgrade bottom in the Z direction and horizontal direction is zero, the thickness of the subgrade has little influence on the upward propagation of reflective cracks. Therefore, to improve the calculation efficiency, the thickness of the subgrade base in the model will be appropriately reduced. In addition, since the crack length is much smaller than the model size, to facilitate the observation of the crack growth trajectory, the result in the red area in Figure 9 is selected for visual processing and analysis.

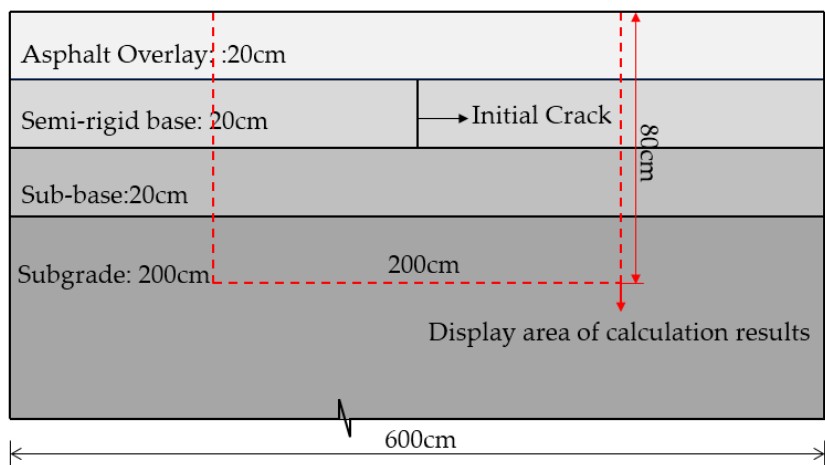

**Figure 9.** Pavement model.

There are many types of cracks in asphalt pavement. The crack propagation behavior in a semi-rigid base is studied in this paper. Figure 9 illustrates the geometry, profile, and thickness of multilayer pavement with initial vertical cracks [14,15]. It is assumed that the initial crack will gradually spread upward under load. Poisson's ratio is taken as 1/3 [28]. The mechanical parameters [33] of the PD model of the pavement structure are shown in Table 3.

**Table 3.** Mechanical parameters of pavement material.

| Pavement Structure | Modulus (MPa) | Poisson's Ratio | Density(kg/m³) | Fracture Energy (J/m²) |
|---|---|---|---|---|
| Asphalt overlay | 3000 | 1/3 | 2300 | 1500 |
| Semi-rigid base | 2800 | 1/3 | 2200 | 980 |
| Subbase | 2200 | 1/3 | 2100 | 980 |
| Soil base | 60 | 1/3 | 1800 | / |

*4.3. Analysis of Vehicle Load Action Form*

When a car runs on a road, the shape of the tire in contact with the road surface is not fixed but varies with the type of vehicle and the roughness of the road surface. To construct a PD model of a pavement structure, optimizing the shape of the contact surface between the tire and asphalt pavement will have a great influence on the simulation results.

The design load of the vehicle in this paper adopts the axle load of a 100 kN single axle–double wheel set stipulated in the specification [39], the tire road surface pressure is 0.70 MPa, the diameter of the equivalent circle of the contact between the single wheel and the road surface is 213.0 mm, and the center distance between the two wheels is 319.5 mm. The road model established in this paper is two-dimensional, so the expression form of vehicle and road structure are both two-dimensional sections. At this point, there will be some errors if the shape of the contact surface between the tire and the road surface is assumed to be circular because the circular shape is simplified into a straight line passing the center of the circle in two dimensions and the shape effect is ignored. To eliminate such errors, the contact surface form of the tire and road surface equivalent circle in the specification is transformed into square contact surface form by referring to the principle of stress equivalence [40,41], and the calculation formula is as follows:

$$\frac{1}{4}\pi R^2 = x^2 \tag{25}$$

where R is the circular diameter of the contact between the single wheel load and the road surface, which is 213 mm, and x is the equivalent length of the square contact surface, which is 189 mm.

Different stress distributions will occur in the asphalt overlay in the process of "close to the crack–above the crack–away from the crack". To simplify the calculation work and reduce the test amount, a load position that causes the maximum damage to the pavement structure is usually selected in the analysis of asphalt overlay composite structures, which is called the critical load position. The PD pavement structure model shown in Figure 10 was established. The thickness of the asphalt overlay was temporarily set as 20 cm, and the material parameters of each layer are shown in Table 3. To simulate the real stress of the pavement structure, the bottom end and the left and right boundaries of the model were fixed. The PD point discrete spacing is Δ = 2 cm, and the horizon δ = 3.0Δ. The two load positions shown in Figure 10 are the situation when the vehicle loads are directly above the crack and on the side of the crack, respectively. When the vehicle load acts on the crack directly above, the corresponding crack tip produces a large compressive stress; when the vehicle load acts on the edge of one side of the crack, the maximum shear stress is generated at the corresponding crack. Therefore, there are two least favorable load conditions, namely, the right above the crack (symmetrical load) and the side of the crack (asymmetrical load).

Under the action of a standard axial load, there is no crack propagation in the asphalt overlay under the positive load and partial load. It can be concluded that the reflective crack is actually caused by the repeated action of load over a long period of time. To explore the crack propagation, the load on the contact between the tire and road surface was increased step by step until crack propagation appeared. After many load-parameter debugging calculations, it is found that when the load increases to 11 MPa, the cracks of

the semi-rigid base of the pavement structure under partial load expand, and the specific situation of crack expansion is shown in Figure 11.

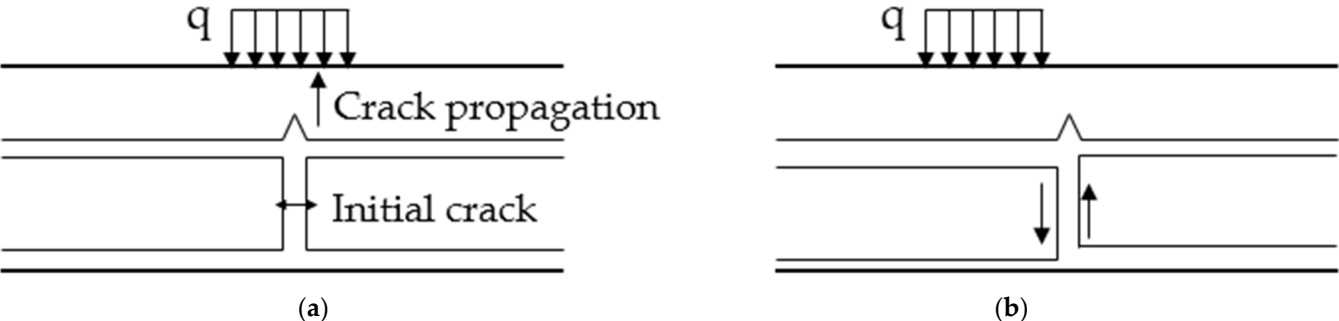

**(a)**
**(b)**

**Figure 10.** Wheel load distribution schematic drawing for different loads. (**a**) Symmetrical load; (**b**) asymmetrical load.

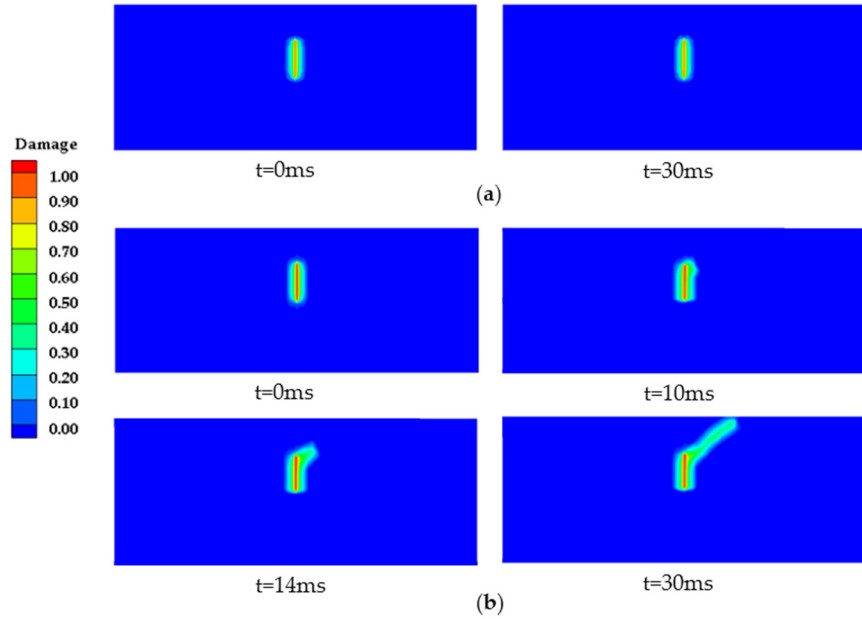

**Figure 11.** Crack propagation under different forms of loads. (**a**) Damage cloud image under symmetrical load; (**b**) damage cloud image under asymmetrical load.

In the process of applying a symmetrical load on the pavement structure until the whole structure reaches stability, the initial cracks in the semi-rigid base do not expand. In the process of asymmetrical loading, the crack tip begins to expand at t = 1 ms. The initial propagation direction of the crack is along the interface between the semi-rigid base and the asphalt overlay. When it is extended for a certain distance, at t = 14 ms, the crack extends upward at a certain angle and runs through the asphalt overlay until t = 40 ms. Due to the existence of the initial cracks in the semi-rigid base, the contact between the base and the asphalt overlay is not stable, and sliding occurs between the adjacent layers, which is why the reflective cracks initially extend horizontally along the interface.

Additionally, it can be seen from Figure 11 that the crack does not expand under symmetric load, but rather under asymmetrical load, which is consistent with the conclusions in [42]. This is because under the action of symmetric load, the shear stress at the crack at the bottom of asphalt overlay is close to zero, and the tensile stress at the bottom of asphalt overlay is negative. At this time, the crack tip is under pressure and the crack closes without expansion. Under the action of asymmetrical load, the tensile stress at the crack is still negative, but the shear stress increases significantly, resulting in shear crack. Therefore,

asymmetrical loading is the main reason that causes the initial cracks of the semi-rigid base to reflect onto the asphalt overlay under the action of the driving load.

### 4.4. Influence of Asphalt Surface Thickness

Several solutions are proposed to retard the reflective cracking problem. Simply increasing the asphalt overlay thickness could be considered the most common retardation treatment. This section will explore the reflective crack propagation under asphalt surfaces of different thicknesses to evaluate the retardation effect of increasing asphalt surface thickness on reflective cracks.

A more refined model is needed to better observe the propagation of reflective cracks. This means that the spacing of PD points needs to be reduced, and at the same time, more PD points are needed, which will greatly increase the calculation time. To make the calculation model more refined without slowing down the calculation efficiency, it is necessary to optimize the road surface model. Generally, when studying the behavior of reflective crack propagation, only the process of crack propagation upward to the asphalt surface is considered, assuming that the base is not cracked. Therefore, this section only considers the interaction between the asphalt surface and semi-rigid base as the main factor, while the influence of the plain base and soil base as the secondary factor will not be considered.

The double-layer pavement structure model shown in Figure 10 was established. The length of the model was 1 m, and the thickness of the semi-rigid base was 15 cm. The bottom of the model was fixed, and both ends were free. The asymmetrical load was selected as the loading mode, and the load was 10 MPa. The PD point discrete spacing is $\Delta = 1$ cm, and the horizon $\delta = 3.0\Delta$. This paper used 3 cm as the gradient of asphalt surface thickness to explore the propagation of reflective cracks on the road surface under different thicknesses of 6–15 cm, and there were four working conditions in total. To better observe the crack propagation trajectory, only the damage cloud map of the crack growth area was selected for display, which is the same as in Section 4.5. Numerical simulation results of the reflective crack propagation of asphalt pavement with different thicknesses under asymmetrical loading are shown in Figure 12.

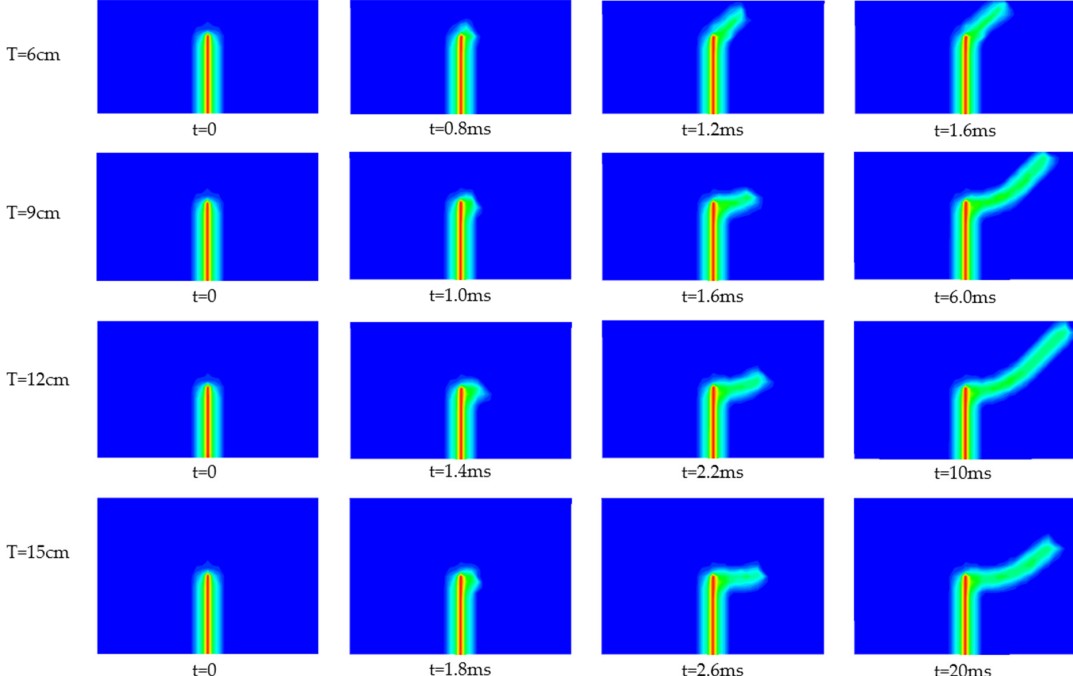

**Figure 12.** Crack propagation of asphalt pavement with different thicknesses.

The expansion behavior of reflective cracks in asphalt overlays of different thicknesses is analyzed in Table 4.

**Table 4.** Expansion behavior of reflective cracks in asphalt overlays of different thicknesses.

| Thickness of Asphalt Overlay | Expansion Behavior of Reflective Cracks |
|---|---|
| T = 6 cm | At t = 0.8 ms, the initial cracks in the semi-rigid base begin to expand and spread along the upward sloping direction, and run through the asphalt overlay at t = 1.6 ms. |
| T = 9 cm | At t = 1.0 ms, the initial cracks begin to extend horizontally along the interface between semi-rigid base and asphalt overlay. At t = 1.6 ms, the fracture propagation direction changes and begins to extend upward at a certain angle to the base. At t = 6.0 ms, cracks run through the asphalt overlay. |
| T = 12 cm | At t = 1.4 ms, the initial cracks begin to extend horizontally. It changes and begins to expand along the upward sloping direction at t = 2.2 ms. At t = 10 ms, cracks run through the asphalt overlay. |
| T = 15 cm | At t = 1.8 ms, the initial cracks begin to extend horizontally. It changes and begins to expand along the upward sloping direction at t = 2.6 ms. At t = 20 ms, the structure is stable and the crack does not run through the asphalt overlay. |

According to the above simulation results, with the increase in asphalt overlay thickness, the time of initial crack initiation is delayed, and the crack propagation speed slows down. In the numerical simulation example of the asphalt overlay with a thickness of 15 cm, the cracks do not penetrate the asphalt overlay when the structure reaches stability. All the above results indicate that increasing the thickness of the overlay has a certain inhibitory effect on the reflective crack propagation. However, increasing the thickness of the asphalt surface means a huge increase in cost, which is not wise. In addition, cracks in the semi-rigid base initially extend horizontally along the interface of adjacent layers, so maintaining a good bond between the semi-rigid base and asphalt surface will also help to slow down the expansion of reflective cracks.

*4.5. Influence of Friction*

Since the friction between the vehicle and road surface is much smaller than that of the vehicle load, most scholars ignore the friction between the wheel and road surface when studying the influence of vehicle load on reflective crack propagation. Without considering the friction between wheels and pavement, due to the symmetry of the structure, when studying the most unfavorable position of load, the two cases of vehicles approaching cracks and vehicles far from cracks can be simplified to analyze the asymmetrical load simultaneously. Once friction is taken into account, the two cases are no longer equivalent (Figure 13). In this section, the friction between the tire and pavement is considered to analyze the crack propagation under the action of two different asymmetrical loads: approaching the crack and far from the crack.

The numerical simulation model was selected as the case of an asphalt overlay with a thickness of 15 cm under asymmetrical loading in Section 4.4, and the material parameter settings of the model are the same as those in Section 4.4. The friction coefficient between the tire and pavement was selected as 0.1 [15], and the direction of friction is opposite to the direction of vehicle operation. To accurately explore the effect of friction between the tire and pavement on reflective crack propagation, a group of frictionless examples were added as the control group. Three groups of different working conditions are set in this section. In case (a), the friction between the wheels and road surface is ignored, and only the vertical vehicle load is considered. In case (b) and case (c), the friction between the tire

and pavement is considered, and the direction of friction in both working conditions is always left. However, the position of the vehicle load changes: one is on the left side of the crack, and the other is on the right. In case (b), the vehicle approaches the crack, and the vertical asymmetric load is to the left of the crack. According to Section 4.4, the crack has a tendency to expand to the right. At this point, the direction of friction is opposite to the trend of crack propagation. In case (c), the vehicle is away from the crack, the vertical asymmetric load is on the right side of the crack, the crack has a tendency to expand to the left, and the direction of friction is the same as the trend of crack propagation. Numerical simulation results of the crack propagation trajectory of the three groups of calculation models at the same time are shown in Figure 14.

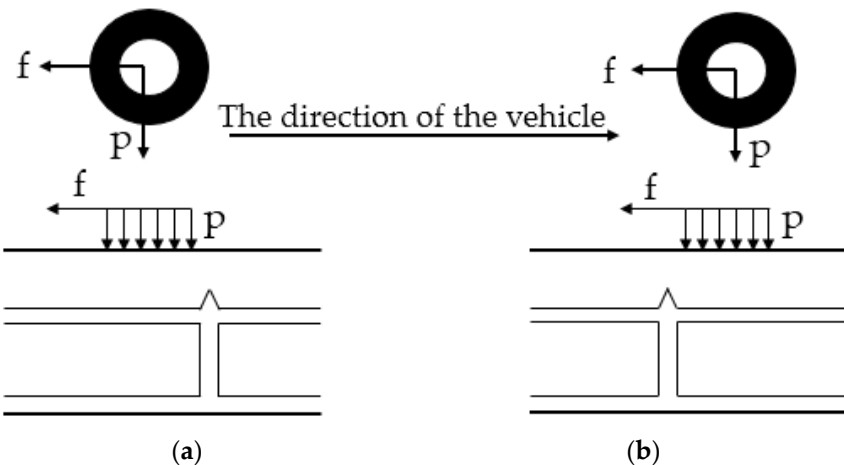

(**a**)                                              (**b**)

**Figure 13.** Schematic diagram of vehicle load action. (**a**) Approaching the crack; (**b**) away from the crack.

In the numerical simulation case (a), the initial crack begins to extend horizontally at t = 1.8 ms. It begins to expand along the upward sloping direction at t = 2.6 ms. At t = 16 ms, the structure is stable, and the crack does not run through the asphalt overlay. In case (b), the crack does not expand, indicating that the friction force in this case has a restraining effect on the crack growth. In case (c), the crack begins to expand at t = 1.4 ms and then begins to tilt upward at t = 2.2 ms until the crack runs through the asphalt overlay at t = 7.6 ms. The crack propagation time is advanced, the crack propagation speed is accelerated, and the cracking degree of the pavement structure is more serious, indicating that the friction force in this case accelerates crack propagation, which is more unfavorable for pavement structures.

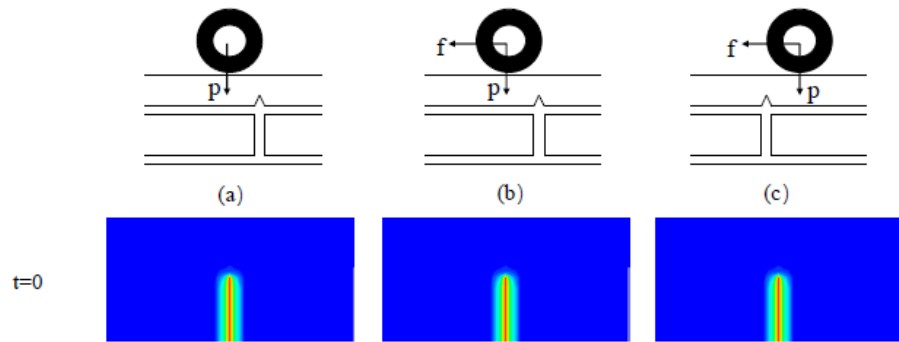

**Figure 14.** *Cont.*

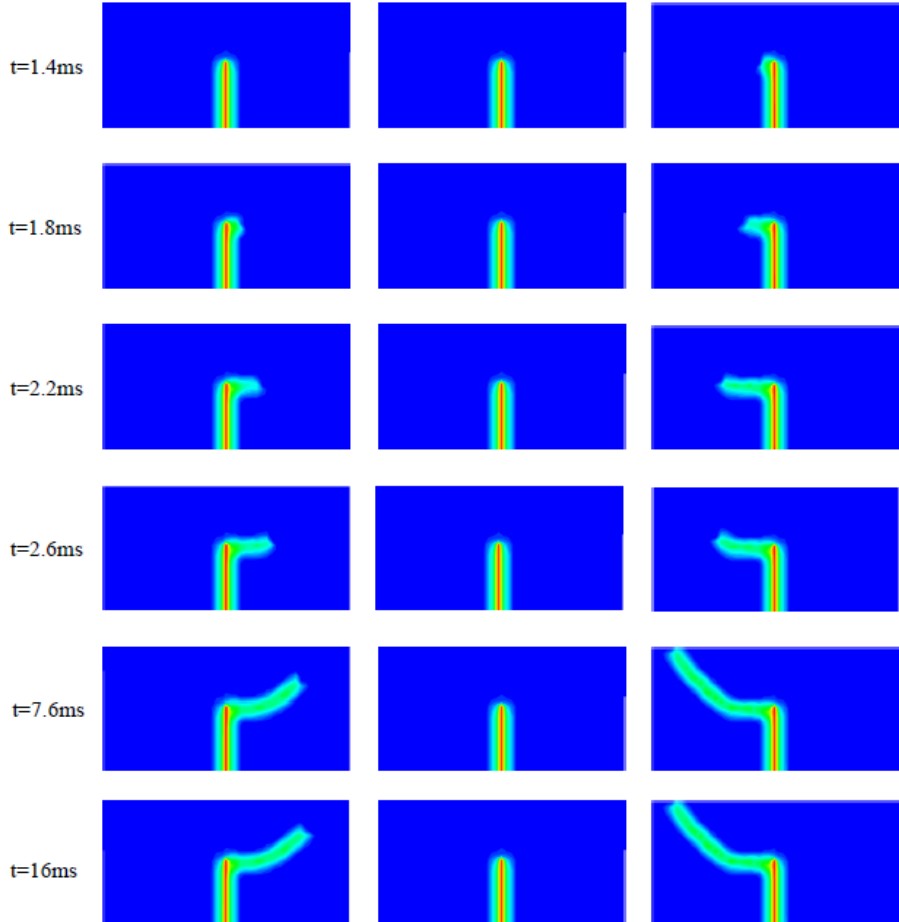

**Figure 14.** Crack propagation of asphalt pavement under the action of wheel friction is considered. (**a**) Vehicle load approaching the crack without friction; (**b**) vehicle load approaching the crack with friction; (**c**) vehicle load away from the crack with friction.

## 5. Conclusions

In this paper, the theory of peridynamics was first modified and verified by three typical examples, and then the method was applied to investigate the crack propagation process of semi-rigid asphalt pavement under load from the three impacts: different forms of load, different thicknesses of asphalt surface, and wheel friction. The main conclusions are as follows:

1. A two-dimensional asphalt pavement delamination model was established by modifying the theory of PD. The simulation results of the proposed model-based PD theory are in good agreement with the analytical solution, experimental results, and finite element calculation results. The feasibility of the application of the model to asphalt pavement structures was verified. It provided a new analytical idea for simulating reflective crack propagation.
2. The propagation of original cracks in the base under different forms of load was compared. The results show that the influence of asymmetric loading on crack propagation is greater than that of symmetric loading.
3. The crack propagation features of asphalt surfaces with different thicknesses were summarized through PD simulation results. That is, the increase in the thickness of the asphalt overlay can delay the time of initial crack propagation, slow the crack propagation speed, and weaken the damage degree of the pavement structure. In addition, maintaining a good bond between the semi-rigid base and asphalt overlay can slow the propagation of reflective cracks.

4.　The influence of friction between tires and roads on reflective crack propagation cannot be ignored. If the friction direction is consistent with the crack propagation direction, the crack propagation will be accelerated. In contrast, crack propagation will be inhibited. The most unfavorable load position is the asymmetrical load position when the vehicle is far from the crack.

**Author Contributions:** Conceptualization, Z.S. and L.X.; methodology, L.X.; software, Z.S.; validation, J.Y., L.X. and Z.S.; formal analysis, Z.S.; investigation, L.X.; resources, X.W.; data curation, Z.S.; writing—original draft preparation, Z.S.; writing—review and editing, L.X.; visualization, Z.S.; supervision, J.Y.; project administration, X.W.; funding acquisition, X.W. All authors have read and agreed to the published version of the manuscript.

**Funding:** This research was funded by the Project of Science and Technology of Henan Transportation Department (Grant number 2019J1).

**Institutional Review Board Statement:** Not applicable.

**Informed Consent Statement:** Not applicable.

**Data Availability Statement:** Not applicable.

**Conflicts of Interest:** The authors declare no conflict of interest.

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
