# Peer review of "Peridynamics for Fracture Analysis of Reflective Cracks in Semi-Rigid Base Asphalt Pavement"

_applsci, doi:10.3390/app12073486_

Round 1

Reviewer 1 Report

This paper establishes a model by Peri-dynamics to analyze the reflective crack in semi-rigid base asphalt pavement. It would be a good idea that taking the model built in this paper to evaluate crack propagation. But some advantages about the model comparing others could be added. And more detailed problems of this paper are raised as follows:

  1. 1. Line 27 to 28: expression problem, “…to changes in temperature and humidity” should be “…to changes in humidity and temperature”, in order to be consistent with “…dry shrinkage cracks and low-temperature shrinkage cracks…”
  2. 2. Line 53: grammar problem, “…during use” should be “…during the use”
  3. 3. Line 130: careless problem, “üis the acceleration…” should be “ü is the acceleration…”
  4. Line 252 to 255: model simulates the pavement structure, so does the data of model correspond to the pavement structure?
  5. 5. Line 270 to 271: it may be more convenient to compare the differences between Figure 7c and Figure 7d if the legends of both are same.
  6. 6. Line283 to 284: references could be cited here to strengthen the view that the application of PD theory to the simulation of low-temperature cracking of asphalt pavement has a scientific, feasible and sufficient research basis.
  7. 7. Line 306: it would be more beautiful after adjusting the text form of the coordinate axes in Figure 8, if possible, the units could be added.
  8. 8. Line 346: does the thickness of each layer given by the model in Figure 9 simulate the thickness of the real pavement structure layer?
  9. 9. Line 372 to 375: in real pavement structure, the bottom end and the left and right boundaries could be viewed as infinite, so in this model, fixing a small size of the model may be not suitable.
  10. 1 Line 410: careless problem, improper use of quotation marks.
  11. 1 Line 410 to 417: it may be not proper to quote such long sentences.
  12. Line 430: more assumptions for model, less authenticity to the analysis results, if base was cracked, how about the result?
  13. 1 Line 437: why the selected load is 10 MPa? It would be better if there is explanation about it.
  14. 1 Line 478: the selected friction coefficient is 0.1, is this coefficient nearer to the real value?
  15. 1 Line 475 to 477, 479 to 481, 485 to 486, in my view, there are two test with same conditions, but the results of both are opposite, it would be better if explanation is added.

Considering above-mentioned reasons, I suggest this paper needs a major revision.

Reviewer 2 Report

Please find attached a word file containing my comments and remarks.

Round 2

Reviewer 1 Report

It can be accepted after minor english editing. 

Reviewer 2 Report

Thank you for the corrections and your explanations. I think the update manuscript has been sufficiently improved to warrant publication in Applied Sciences.